# Dietary Modulation of the Human Gut Microbiota and Metabolome with Flaxseed Preparations

**DOI:** 10.3390/ijms231810473

**Published:** 2022-09-09

**Authors:** Karin Kleigrewe, Martina Haack, Martine Baudin, Thomas Ménabréaz, Julien Crovadore, Mahmoud Masri, Michael Beyrer, Wilfried Andlauer, François Lefort, Corinna Dawid, Thomas B. Brück, Wolfram M. Brück

**Affiliations:** 1Bavarian Center for Biomolecular Mass Spectrometry, School of Life Sciences, Technical University of Munich, 85354 Freising, Germany; 2Werner Siemens-Chair of Synthetic Biotechnology, Department of Chemistry, Technical University of Munich, Garching b., 85748 München, Germany; 3Institute of Life Technologies, School of Engineering, HES-SO University of Applied Sciences and Arts Western Switzerland Valais-Wallis, 1950 Sion, Switzerland; 4Plants and Pathogens Group, Research Institute Land Nature and Environment, Geneva School of Engineering, Architecture and Landscape (HEPIA), HES-SO University of Applied Sciences and Arts Western Switzerland, 1254 Jussy, Switzerland; 5Chair of Food Chemistry and Molecular Sensory Science, School of Life Sciences, Technical University of Munich, 85354 Freising, Germany

**Keywords:** flaxseed, plant bioactives, microbiota, enterolignans, cyclolinopeptides

## Abstract

Flaxseeds are typically consumed either as whole flaxseed, ground flaxseed, flaxseed oil, partially defatted flaxseed meal, or as a milk alternative. They are considered a rich source of vitamins, minerals, proteins and peptides, lipids, carbohydrates, lignans, and dietary fiber, which have shown hypolipidemic, antiatherogenic, anticholesterolemic, and anti-inflammatory property activity. Here, an in vitro batch culture model was used to investigate the influence of whole milled flaxseed and partially defatted milled flaxseed press cake on the gut microbiota and the liberation of flaxseed bioactives. Microbial communities were profiled using 16S rRNA gene-based high-throughput sequencing with targeted mass spectrometry measuring lignan, cyclolinopeptide, and bile acid content and HPLC for short-chain fatty acid profiles. Flaxseed supplementation decreased gut microbiota richness with Firmicutes, Proteobacteria, and Bacteroidetes becoming the predominant phyla. Secoisolariciresinol, enterodiol, and enterolactone were rapidly produced with acetic acid, butyric acid, and propionic acid being the predominant acids after 24 h of fermentation. The flaxseed press cake and whole flaxseed were equivalent in microbiota changes and functionality. However, press cake may be superior as a functional additive in a variety of foods in terms of consumer acceptance as it would be more resistant to oxidative changes.

## 1. Introduction

Flax or linseed (*Linum usitatissimum* L.) belongs to the family Linaceae and is native to Western Asia, North Africa, and Southern Europe. Today, flax is grown in about 50 countries, most of which are in the northern hemisphere [1]. With circa 40% of the world’s production, Canada is the largest producer of flaxseed to date [2]. In general, flaxseed contains up to 50% oil. Flaxseed meal comprises 23% to 34% protein, 4% ash, 5% viscous fiber, and lignan precursors (9 to 30 mg/g of defatted meal) [3]. Traditionally used for their fibers in the textile industry, flaxseeds have now been widely adopted in the food industry. Products from flax for human consumption include whole flaxseed, ground flaxseed, flaxseed oil, partially defatted flaxseed meal and flaxseed milk alternatives [4,5]. Flaxseeds also contain antinutrients, but there is no indication that flaxseed could cause toxicity in humans [4,6]. Hence, flaxseed is generally recognized as safe (GRAS) and is considered a rich source of vitamins, minerals, proteins and peptides, lipids, carbohydrates, lignans, and dietary fiber. Bioactive compounds from flaxseed have shown to have hypolipidemic, antiatherogenic, anticholesterolemic, and anti-inflammatory properties, amongst others [4,5,6,7,8,9]. Due to its high content of alpha linolenic acid (ALA), flaxseed is susceptible to oxidation, causing rancidness, off flavors, bitterness, and a musty aroma [4,8]. This bitterness of linseed and, particularly, linseed oil is due in part to a group of cyclic octa- and nonapeptides, called cyclolinopeptides. The latter, furthermore, may have anti-inflammatory and hepatoprotective effects [9].

In addition to n-3 fatty acids, phenolic compounds, such as lignans and dietary fibers, make up a significant fraction of flax composition. Flaxseeds contain insoluble and soluble fibers in a proportion ranging from 60:40 to 80:20 [10]. The major insoluble fiber fraction consists of cellulose and lignin [11]. The main soluble fiber from flaxseed has been reported to be a heterogeneous polysaccharide consisting of neutral arabinoxylan and acidic rhamnose [12]. This flaxseed polysaccharide exhibited good antioxidant activities in vitro and reduced elevated blood glucose in rats [13]. Despite their physiological advantages, little research has been conducted on the fermentability of flaxseed fibers. A recent study by Arora et al. [14] found that flaxseed fibers promoted the cecal proliferation of *Bifidobacterium* and *Akkermansia* and were fermented into short-chain fatty acids (SCFAs) and other small organic acids in mice models. The production of acetic acid from flaxseed xyloglucans and butyric acid from flaxseed arabinoxylans was significantly higher than in negative controls following the 48 h in vitro fermentation of pig colonic digesta [15].

Dietary lignans have been linked to several health benefits such as protection from osteoporosis, cardiac and liver disease, and the reduction of plasma cholesterol [16]. As such, lignans have been recognized as nutraceuticals and functional foods [17]. In plants, lignans exist predominately as glycosides [18]. Flaxseeds contain 75 to 800 times more lignans compared to other cereal crops [6,7]. The most abundant lignin is secoisolariciresinol diglucoside (SDG) and its deglycosylated form, secoisolariciresinol (SECO) [19]. Flaxseed contains the highest concentrations of SECO of any food (28–369 mg/100 g) [18]. Flaxseed also contains other lignans such as matairesinol (1100 μg/100 g of full fat flaxseed), pinoresinol (PINO) (870 μg/411 g), lariciresinol (LA) (1780 μg/100 g), pinoresinol diglucoside (PDG), medioresinol, isolariciresinol, syringaresinol, and hydroxymatairesinol [7,20,21].

SDG and SECO have been shown to be of particular interest as they are converted in mammalian colon to the enterolignans (mammalian lignans) enterodiol (ED) and enterolactone (EL) [21]. Enterolignans may possess a variety of biological activities such as antioxidant activity and may influence cellular pathways that are important to cancer risk [22]. They may also reduce the plasma levels of free estradiol and testosterone, and thus, affect the development of hormone-dependent diseases [23]. EL inhibited the proliferation of estradiol-stimulated MCF-7 human breast cancer cells in vitro by competing for sulfokinases and sulfatases involved in estrogen metabolism [24].

The production of gut bacteria plays an important role in the formation of enterolignans as plant lignan glycosides are poorly absorbed by the small intestine and resist hydrolysis by mammalian enzymes [4]. The transformation of SDG in the mammalian colon is performed by the resident anaerobic microbiota. Firstly, glucose moieties are removed to produce an intermediate, secoisolariciresinol monoglucoside (SMG) before being further deglycosylated into SECO. SECO is then demethylated and decarboxylated to yield ED and EL [25,26]. SDG hydrolysis may be performed by bacteria possessing β-glucosidase (β-glu, EC 3.2.1.21) activity, which include strains of *Bacteroides*, *Clostridium*, and some *Bifidobacterium* [27]. As a result, the probiotic bifidobacteria are of interest to promote the production of ED and EL in the gut, as they already are implicated in a variety of health-promoting benefits. Unfortunately, it has been found that SDG conversion is not a common feature of *Bifidobacterium*. In pure cultures, out of 28 strains, only *Bifidobacterium pseudocatenulatum* WC 401 showed a 75% conversion of SDG to SECO after 48 h in a cellobiose-based medium [27]. In other studies, EL production was linked to an abundance of *Ruminococcus* species, particularly, *R. bromii* and *R. lactaris,* as well as *Lactobacillus-Enterococcus* and *Methanobervibacter* [28]. *Bilophila* were inversely correlated with EL and ED production [29]. The composition of dominant fecal bacterial (Bacteroidetes and Firmicutes) communities was not significantly affected by flaxseed ingestion at nearly all taxonomic levels analyzed [28]. Similarly, a pilot study using lignan-rich oilseeds in in vitro batch culture fermentations of fecal microbiota of healthy younger and premenopausal females pointed towards *Klebsiella* and *Collinsella* spp. as prospective actors in EL and ED production. Further observations showed that 24 h of in vitro batch culture fermentation with the oilseed mix resulted in decreased proportions of Ruminococcaceae and members of Lachnospiraceae, whereas Enterobacteriaceae increased significantly (*p*  ≤  0.1) [29]. This was also observed in a study supplementing the diet of Peking ducks with flaxseed. Here, flaxseed-fed groups had significantly higher abundances of *Escherichia*, *Shigella*, and *Campylobacter* (*p* < 0.1). Flaxseed supplementation in the same study also changed the abundance of pro-inflammatory bacteria: Veillonellaceae first increased and then decreased significantly (both *p* < 0.05), which was associated with serum prostaglandin E2 and leukotriene B4 [30].

In this study, we used an in vitro batch culture model to investigate the dynamics of the human gut microbiota stimulated by defatted flaxseed press cake and whole flaxseed supplementation. In particular, flaxseed lignans, enterolignans, the gut microbiota, cyclolinopeptides, and metabolic end products of microbial fermentation are examined. These ex vivo models facilitate the preliminary screening of microbiome modulators and have proven their use and validity in a large number of previous studies [31]. Whole milled flaxseed and partially defatted milled flaxseed press cake were used. Microbial communities were profiled using 16S rRNA gene-based high-throughput sequencing with targeted mass spectrometry measuring lignan, cyclolinopeptide, and bile acid content and HPLC for short-chain fatty acid profiles. The in vitro batch system is a host-free platform that can simulate the spatial, temporal, and environmental features that microbes encounter in the gut lumen. They are ideal systems for studying microbial perturbations resulting from the addition of exogenous stimuli.

## 2. Results

### 2.1. Characterization of Chemically and Enzymatically Predigested Flaxseed Press Cake and Milled Whole Flaxseeds

#### Sample Hydrolysis, Extraction, and Quantification

After the two-step in vitro digestions mimicking the role of stomach and pancreas, particle sizes of heterocolloids from defatted flaxseed press cake and milled, whole flaxseeds had between a 350 and400 µm diam. After digestion, fine, slightly brown powders were obtained: 45 g from digested flaxseed press cake and 35 g from digested milled whole flaxseed. The fiber content of the digested flaxseed press cake was 43.10 g × 100 g mass-1 (insoluble dietary fiber) and 7.44 g × 100 g mass-1 (soluble dietary fiber). The fiber content of the digested milled whole flaxseeds was 40.90 g × 100 g mass-1 (insoluble dietary fiber) and 7.38 g × 100 g mass-1 (soluble dietary fiber).

### 2.2. Impact of Flaxseed on Fecal-Derived Microbial Communities

Next, the predigested flaxseed preparation was incubated with human gut microbiota of three different donors. The analysis of the impact of the flaxseed preparation on the luminal content of the adult human gut microbiota was assessed using a batch culture, followed by sequencing of V3 and V4 regions of the 16S rRNA gene using an Illumina MiSeq. We obtained 1,7045,399 amplicon raw sequences for 27 fecal samples that we filtered by length, quality, and chimera, giving a total of 1,4473,113 filtered sequences with an average of 24,971 sequences per sample. Alpha diversity analysis showed that control fermentations maintained stable populations (Table 1). However, both fermentations using digested, defatted flaxseed press cake and digested whole flaxseed had perturbed populations from 6 h of fermentation onwards compared to control fermentations (*p* < 0.05). To assess the overall bacterial composition for the control and the whole flaxseed and flaxseed press cake groups, we performed a generalized UniFrac.

Analysis using a dendrogram and non-metric multidimensional scaling plot (NMDS) showed good separation of control fermentations without flaxseed at 6 h, 18 h, and 24 h, which clustered the point of inoculation (T0) from the test fermentations using digested, defatted flaxseed press cake and digested whole flaxseed (Figure 1a,b).

Batch cultures using digested flaxseed press cake and digested milled whole flaxseed clustered together in their respective time points. A stacked bar plot of the relative abundance of the 15 most predominant genera showed that Bacteroides OTUs were the most relative abundant taxa at the point of inoculation (T0) and in all the control fermentations (T6 C, T18 C, T24 C), followed by Faecalibacterium, Roseburia and Parabacteroides OTUs (Figure 2). In fermentations using digested, defatted flaxseed press cake and digested whole flaxseed populations at 6 h of fermentation (T6 press cake (PC) and T6 whole flaxseed (FS)), Escherichia-Shigella OTUs were the most relative abundant taxa (*p* < 0.05), followed by Bacteroides. From T6 onwards, populations also showed an increase in the relative abundance of Acidaminococcus (*p* < 0.05), Dialister (*p* < 0.05), and Parasutterella OTUs. The relative abundance of Bacteroides, Faecalibacterium, and Eubacterium OTUs (Eubacterium eligens group, Eubacterium ventriosum group, Eubacterium xylanophilum group) decreased. At time points 18 h and 24 h, Acidaminococcus became the OTU with the most relative abundance in fermentations using digested, defatted flaxseed press cake (*p* < 0.05) and digested whole flaxseed populations (*p* < 0.05) (T18 PC, T24 PC, T18 FS, T24 FS). The relative abundance of Veillonella (*p* < 0.05) and Lachnospira (*p* < 0.05) OTUs (Lachnospira, Lachnospiraceae-ND3007-group, Lachnospiraceae-NK4A136-group, Lachnospiraceae-UCG-001, Lachnospiraceae-UCG-004) also increased significantly, whereas the relative abundance of Escherichia-Shigella OTUs decreased but did not return to levels compared to the inoculum or the control fermentations. Genera with a known effect on EL production only represented a minor part of the overall population with relative abundances not exceeding 1.5% at most. Notably, Akkermansia, Bifidobacterium, and Collinsella generally increased during fermentation as compared to the inoculum, whereas Ruminococcaceae (Ruminococcaceae NK4A214 group) decreased and Lachnospira remained stable at inoculum levels. Bifidobacterium spp. initially increased in the initial 6 h of fermentation (from 0.15% to 0.52% and 0.93% in press cake and whole flaxseed, respectively). Between 6 h and 24 h, the relative abundance of bifidobacteria, however, decreased again (0.33% and 0.36% in press cake and whole flaxseed, respectively).

Similarly, Akkermansia was present at 0.01% at T0, then increased to 0.10 and 0.34 in press cake and whole flaxseed, respectively. The abundance of Collinsiella increased in the initial 6 h of fermentation and then remained stable (from 0.04% at T0 to 0.28% and 0.40% at T6). Genera belonging to the Ruminococcaceae were initially present at 0.24%, but disappeared after 18 h of fermentation in both flaxseed fermentations. Similar trends were detected in control fermentations without flaxseed and showed the populations of Lachnospira slightly increased to 1.29% after 24 h of fermentation. Genera representing the Ruminococcaceae also decreased but did not disappear completely.

Relative abundances at the phylum level within the bacterial communities of the different fermentations showed that Bacteroidetes was the most abundant at T0 and in the control fermentations (T6 C, T18 C, T24 C). In fermentations using digested, defatted flaxseed press cake and digested whole flaxseed populations at 6 h of fermentation (T6 PC and T6 FS), Proteobacteria became the most abundant phylum (55.67% and 42.13%, respectively). In fermentations using digested, defatted flaxseed press cake and digested whole flaxseed 18 h and 24 h after inoculation (T18 PC, T24 PC, T18 FS, T24 FS), Firmicutes represented the most relatively abundant phylum with values between 56.26 and 64.01% of classified reads per sample. The Bacteroidetes phylum then came second in terms of relative abundance (Figure 3).

### 2.3. Analysis of Fiber-Derived Short-Chain Fatty Acids in Fermented Samples

In batch cultures using both linseed preparations, acetic, butyric, propionic, succinic, and lactic acids accumulated over the 24 h time period of the incubation of the batch culture (Figure 4). Acetic acid was predominant and accumulated from 0.19 (±0.002) mg/mL at T0 to 4.05 (±0.582) mg/mL and 4.09 (±0.445) mg/mL at T24 in press cake (*p* < 0.05) and ground whole flaxseed (*p* < 0.05), respectively. Butyric acid accumulated from 0.03 (±0.014) mg/mL at T0 to 1.62 (±0.396) mg/mL at T24 in press cake (*p* < 0.05) and 1.44 (±0.622) mg/mL at T24 in ground whole flaxseed (*p* < 0.05). Propionic acid accumulated from 0.01 (±0.005) mg/mL at T0 to 0.98 (±0.189) mg/mL and 1.04 (±0.256) mg/mL at T24 in press cake and ground whole flaxseed, respectively. Lactic acid increased between inoculation (T0) and 6 h of fermentation (T6). However, the lactic acid concentration then decreased until the end of the fermentation after 24 h. Other acids occurred only sporadically in trace amounts. In control fermentations without flaxseed, all examined acids increased throughout the fermentation.

### 2.4. Analysis of Lignans, Cyclolinopeptides, and Bile Acids in Fermented Samples

#### 2.4.1. Lignans

The diglucosides SDG and PDG are rapidly degraded by the bacteria, which consequently increase the concentrations of SECO and PINO within 18 h and 6 h of fermentation, respectively. PINO is also degraded and is not detected in later fermentation time points, whereas SECO is detected at later time points. LA is present at T0 and T6 and is then degraded as well. As described in the literature, ED and EL are formed out of these lignan precursors (Figure 5).

#### 2.4.2. Cyclolinopeptides

Quantitative analysis of different flaxseeds showed that CL1, 1-Met-CL2, 1-Met-CL3, 1-Met-CL4, and 1-Met,3-Met-CL6 (Lang et al. 2022) are the most abundant cyclolinopeptides present in fresh flaxseed (Figure 6). During storage, the methionine group is oxidized, forming methionine sulfoxide (Mso) or the methionine sulfone (Msn). The quantitative analysis of the cyclolinopeptides present in PC and FS demonstrated that during the preparation of the PC and FS, oxidation must have occurred, showing the presence of 1-Mso-CL3, 1-Mso-CL4, and 1-Mso-CL2. CL1’s presence is due to its chemical structure not being prone to oxidation. Interestingly, during the fermentation, the concentration of the non-oxidized CL, such as 1-Met-CL4 and 1-Met-CL2, increased, showing the reductive potential of the fermentation culture.

#### 2.4.3. Bile Acids

The fermentation culture was supplemented with bile salts #3 which mainly consisted of cholic acid (CA) and deoxycholic acid (DCA) (Figure 7). At T0, not only are the primary bile CA and DCA present, but the bile acids are as well; the bile acids are conjugated with glycine or taurine. During the fermentation process, the bile acids were depleted, forming the secondary bile acids 7-ketolithocholic acid (7-KLCA) and α-muricholic acid (α-MCA).

## 3. Discussion

Flaxseed and its bioactive components have been implicated in several health benefits, positively impacting cardiovascular disease, diabetes, hormonal status, brain health, skin health, gut health, and cancer [4]. Most of these effects were due to flaxseed containing alpha-linolenic acid (ALA), lignans, and fiber [5]. Flaxseed cyclolinopeptides were found to have anti-inflammatory and hepatoprotective effects [9]. The objective of this study was to examine the compositional changes of the human gut microbiota and follow its metabolic impact in batch cultures with defatted flaxseed press cake and whole flaxseed supplementation.

### 3.1. Microbial Communities

Fermentations using either whole flaxseed or defatted flaxseed press cake showed almost identical microbial profiles, which significantly (*p* < 0.05) differed from the control fermentation and the inoculum. Flaxseed supplementation decreased the richness of gut microbiota when compared to control fermentations. Firmicutes, Proteobacteria, and Bacteroidetes became the predominant phyla. This conflicted with most previous studies, but coincided with the findings of Wu et al. and Xia et al. [30,32]. After 6 h of fermentation, Proteobacteria belonging to the genera *Escherichia* and *Shigella* became dominant with significant concomitant increases in *Veillonellaceae* and *Dialister*. This is congruent with earlier reports that demonstrated that the genera *Escherichia* and *Shigella* increased in flaxseed-fed groups alongside pro-inflammatory bacteria, including members of the taxa *Negativicutes*, *Veillonellaceae*, *Dialister*, *Megamonas*, and *Prevotella* [30]. In contrast, anti-inflammatory bacteria, including *Clostridia*, *Porphyromonadaceae*, *Lachnospiraceae*, and *Bacteroides* diminished, but increased again with a prolonged feeding duration [30,32]. *Faecalibacterium* and *Akkermansia*, two important commensal bacteria of the human gut microbiota, also decreased. Experiments in animal models using flaxseed showed that flaxseed supplementation reduced *Akkermansia muciniphila* abundance. Furthermore, flaxseed mucilage showed a decrease in *Faecalibacterium* [33,34]. After 18 h, Firmicutes and the genus *Acidaminococcus* showed the highest relative abundance. Increases in the genus *Acidaminococcus*, belonging to the phylum Firmicutes, has been associated with acute myocardial infarction and the flaxseed oil supplementation of lambs’ diet [35,36,37].

### 3.2. Short-Chain Fatty Acids

In this study, acetic acid was the predominant SCFA, followed by butyric and propionic acids in both press cake and whole flaxseed samples. Acetate may be converted to acetyl-CoA and incorporated in the tricarboxylic acid (TCA) cycle. Within the gut, acetic acid may improve the insulin resistance in obese individuals by affecting peripheral tissues that collectively improve body weight control and insulin sensitivity [38]. Propionic acid and butyrate are resorbed by the intestinal epithelium and are an important energy source used for the growth and maintenance of colonocytes. Furthermore, SCFAs also enhance gut barrier integrity while modulating inflammation-related signaling pathways [39]. SCFAs are also suspected to impact glucose homeostasis and lipid metabolism by affecting distinct hormones involved in appetite regulation [40]. It has been previously observed in in vivo studies that flaxseed increased the production of the SCFA acetic acid, propionic acid, and butyric acid [41,42]. In these studies, acetic acid was also the predominant SCFA as observed here, reaching 80% of total SCFA after 48 h in whole flaxseed and 97% of total SCFA after 48 h in flaxseed flour. Butyric acid decreased over the fermentation period, whereas the content of other acids was not significant [41]. In our study, butyric acid and propionic acid also increased with whole flaxseed and defatted flaxseed press cake samples over a 24 h period, whereas lactic acid increased in the first 6 h of fermentation and then decreased again. Lactic acid may be further metabolized to acetate, propionate, and butyrate by gut microbiota, where both d- and l-lactate were similarly fermented [43]. Amino acid fermentation may also contribute to the production of SCFAs by providing an acidic environment suitable for acidophilic microorganisms. Using amino acids to produce SFCAs mainly yields acetate and propionate [40]. As the present study reports a significant abundance of the genus *Acidaminococcus*, the production of acetic acid, butyric acid, and propionic acid may, in part, be due to free amino acids in the flaxseed fermentations as reported by Jumas-Bilak et al. [44].

### 3.3. Lignans

As previously reported in the literature, the human gut microbiota was able to rapidly degrade SDG into SECO, ED, and EL. Although it has been suggested that the deglycosylation of SDG may occur via the action of brush border enzymes, it appeared from the present study that the gut microbiota is able to perform this step without host intervention [45]. SDG and SECO were further metabolized to the mammalian lignans ED and EL by human intestinal microflora, starting at 18 h of the batch culture. Defatted press cake in this study seemed to be the efficient substrate for the human gut microbiota as a higher concentration of both EL and ED were detected. This has been similarly reported in a previous study comparing whole flaxseed with flaxseed flour [41]. A study comparing whole flaxseed and flaxseed oil suggested that flaxseed fiber supports SDG microbial metabolism to produce EL and ED [46,47]. The production of EL and ED occurred through the demethylation of SECO, forming dihydroxyenterodiol [33]. Organisms implicated in this step were *Butyribacterium methylotrophicum*, *Eubacterium callanderi*, *Eubacterium limosum*, *Ruminococcus productus*, and *Peptostreptococcus productus*. Dehydroxylation by *Clostridium scindens*, *Clostridium saccharogumia*, *Blautia producta*, or *Eggerthella lenta* then resulted in the formation of ED [33]. Roncaglia et al. [27] further attributed EL and ED production to *Bacteroides* spp. and *Bifidobacterium* spp. Dehydrogenation of ED to EL was shown to be accomplished by *Lactonifactor longoviformis* [26]. In the present study, *Bacteroides* spp. and *Eubacterium* spp. were part of the 15 genera with the most relative abundance, whereas *Bifidobacterium* spp., *Blautia* spp., *Clostridium* spp., and *Ruminococcus* spp. were minor components. Hence, it may be speculated that the genera present here were able to perform SDG degradation [48]. However, SECO was also not completely degraded to ED and EL in this study. Hu et al. [49] reported that the antioxidant activity of SECO was higher than ED and EL, which suggests that SECO may diminish some of the negative effects of the pro-inflammatory bacteria that increased during flaxseed fermentation.

This has also been observed with PDG. Although its derivatives PINO and LA were already present in our samples at T0, it was completely degraded after 6 h, leaving only PINO and LA, with PINO being only present in whole flaxseed samples. LA was present in both samples of press cake and whole flaxseed. Likewise, in our results, in vivo studies also showed that LA increased significantly in study subjects and could be formed from PINO, as well as from syringaresinol, following the production of PINO from PDG [50]. The degradation of PINO to LA by demethylation was attributed to *Enterococcus faecalis* and *Peptostreptococcus productus* [26,51]. Degradation by human gut microbiota of LA to ED and EL has been suggested by Xie et al. [51] via a pathway of conversion of LA to SECO. This would explain the disappearance of PDG, PINO, and LA after 6 h of digestion here.

### 3.4. Cyclolinopeptides

The lack of consumer acceptance of flaxseed products is partially due to the presence of cyclolinopeptides that trigger a rapidly developing bitter taste [9]. Here, the quantitative analysis of the cyclolinopeptides present in flaxseed press cake and whole flaxseed demonstrated that during the processing of the in vitro digestion before batch culture fermentation, oxidation must have occurred. During fermentation, cyclolinopeptide profiles shifted towards 1-Met-CL2 and 1-Met-CL4. Although previous studies have suggested that the human gut microbiota interacts with cyclolinopeptides, it remains unknown how and to what extent flaxseed cyclopeptides may affect the human microbiota [52]. Dietary cyclolinopeptides may remain stable in the human gut, which could in turn aid their antimicrobial and antimalarial activities [9]. Further investigations using cyclolinopeptides and pathogenic consortia may be useful in supplementing this field.

### 3.5. Bile Acids

Human gut microbiota associated with fiber metabolism may affect secondary bile acid (BAs) metabolism [53]. Several species of gut bacteria, including *Clostridium*, *Enterococcus*, *Bifidobacterium*, *Lactobacillus*, and *Bacteroides*, can convert primary bile acids into secondary bile acids, such as lithocholic acid (LCA) and deoxycholic acid (DCA) [54]. In vivo, EL has been shown to interact with nuclear receptors involved in bile acid metabolism. Therefore, EL may mediate bile acid synthesis and metabolism [55]. An animal trial assessing the microbiota of C57BL/6 mice on a “Western” high-fat diet (HFD) showed that BAs positively correlated with Firmicutes, Proteobacteria, and Actinobacteria had positive correlations, whereas there was a negative correlation with Verrucomicrobia and TM7. The authors suggested that BAs are the most relevant metabolites affecting gut microbiota [56]. In this study, bile acids were depleted and formed the secondary bile acids 7-KLCA and α-MCA, whereas a microbiota shift towards Firmicutes, Proteobacteria, and Actinobacteria was observed. This suggests, at least partially, that BAs may contribute towards microbiota changes. However, no statistically significant correlations between the production of secondary bile acids from flaxseed preparations and changes in the microbiota composition were observed here.

## 4. Materials and Methods

### 4.1. Flaxseed Processing

Flaxseed, harvested in the year 2017, was obtained from Naturoel AG (Lanzenneunforn, CH). Flaxseed oil was produced by cold pressing (Oil Press FX20, Oeltech Maschinenbau, Munich, Germany) 10 kg of flaxseed. The compression of the press cake was controlled via the manual slit distance regulation, as well as via the oil temperature (50 °C). The oil (2.5 kg) was discarded and the defatted flaxseed press cake (7.5 g) and original whole flaxseed were further processed by dry milling (Pin Mill C100 UPZ, Alpine, Augsburg, Germany) with a 3 mm screen insert at room temperature (air-cooled). Then, 500 g of the two matrices, the defatted press cake and the whole flaxseed, were mixed with 4500 g of demineralized water for wet milling with a ball mill (CoBall-Mill MS 12, FrymaKoruma, Rheinfelden, Switzerland), with 1 mm ceramic beads (SiLibeads Type ZY-E, Sigmund Lindner GmbH, Warmensteinach, Germany), for 10 min at a rotary speed of 10 m/s. The jacket cooling system was operated with tap water. Heterocolloids from milled press cake and milled whole flaxseed were obtained. The mean particle size of these aqueous dispersions was determined with a Camsizer XT (Retsch Technology, Haan, Germany), equipped with a XT-flow module (LOD 5 µm). These heterocolloids were used for the two-step in vitro digestion.

### 4.2. In Vitro Digestion

A two-stage in vitro digestion model based on the procedure described by Xie was used with some modifications [57]. A volume of 650 mL of the linseed heterocolloids was diluted with water (1:1) up to 1300 mL. The gastric phase was initiated by adding 240 mL of 0.05 mol/L HCl solution (pH 1.3) to the heterocolloid solutions. A pepsin solution (130 mL; 3.5 g of pepsin from porcine stomach mucosa in 500 mL HCl 0.1 mol/L) was added, and the mixture was purged with nitrogen and placed in a Multitron PRO incubator (Blanc Labo, Lonay, Switzerland) at 37 °C for 1 h with constant shaking. After the gastric digestion, the pH was adjusted to 6.5 using a NaOH solution (1 mol/L). After the addition of 130 mL of a pancreatin and bile salt solution (3.5 g and 3.7 g, respectively in 500 mL of 0.1 mol/L NaHCO_3_), the mixture was purged with nitrogen and placed at 37 °C for 2 h with constant shaking. During the whole digestion process, the solutions were protected from light.

### 4.3. Processing of Digested Flaxseed Press Cake and Whole Flaxseed

Digested hydrocolloid solutions were centrifuged at 13,000× *g* for 45 min at room temperature. The supernatant was ultrafiltered with a cross flow diafiltration (LabStak M-20, DDS Filtration/Alfa Laval, Lund, Sweden) that was equipped with a stacked membrane module (cut-off = 1000 Da, ETNA 01A, DDS Filtration/Alfa Laval, Denmark) against deionized water with a diafiltration factor of 3. The retentates from the digested press cake and digested original flaxseed were lyophilized with a freeze-drying plant (Sublimator 3 × 4 × 5, Zirbus Technology, Bad Grund, Germany). Samples were frozen at −40 °C on the shelves before the vacuum was set to 0.2 mbar. Then, the shelves’ temperature was ramped up to −5 °C and kept constant until the product temperature reached −5 °C on all shelves. A post-drying step at 20 °C and the maximum power of the vacuum pump finished the drying process. The obtained powders were analyzed for their soluble and insoluble fiber contents.

### 4.4. Soluble and Insoluble Fiber Content

The powders from lyophilization were analyzed using a commercial test kit from Megazyme (Integrated Total Dietary Fiber Assay Kit, Dublin, Ireland), for the determination of insoluble (IDF) and soluble (SDF) dietary fiber (including resistant starch). This method corresponds to the method of the Association of Official Agricultural Chemists (AOAC), Method 2011.25.

### 4.5. Dry Matter of Lyophilized Powders

All the results were expressed per dry matter, which was determined using the HG53 Halogen Moisture Analyzer (Mettler Toledo, Greifensee, Switzerland).

### 4.6. Stool Sample Collection and Preparation

Testing of the digested linseed samples was performed using pooled, freshly voided fecal samples from 3 healthy adult volunteer donors between the ages of 30 and 45 years following a standard Western diet and with no recent history of probiotic, prebiotic, or antibiotic use (within the last 6 months). Information on the donors’ health status, lifestyle habits, clinical anamnesis, and medicine use was collected with a pre-informative questionnaire. For inoculation of the batch culture model, 10% (*w*/*v*) fecal slurries were prepared using anaerobic phosphate-buffered saline (0.1 M, pH 7.2) and homogenized for 2 min at 460 paddle-beats (Stomacher 400, Seward, West Sussex, UK). The maximum time for sample preparation after reception of the fecal sample and inoculation of fermenter vessels was 15 min.

### 4.7. In Vitro Batch Culture Microbiota Model

Triplicate batch culture fermentations were carried out as previously described by Gibson et al. [58]. For the anaerobic batch culture system, a continuous culture growth medium (CMGM, Table 2) was prepared in 1 L of deionized water, adjusted to pH 6.5, and sterilized by autoclaving. Media with linseed preparations contained a sterile CMGM containing 4 mL/L of a 250 mg/L stock solution of resazurin (Fisher Scientific, Reinach, Switzerland), 0.5 g/l L-Cysteine HCl (Sigma, Buchs, Switzerland) and 1% (*w*/*v*) of digested linseed samples. A control fermentation was performed using only a sterile CMGM containing 4 mL/L of a 250 mg/L stock solution of resazurin (Fisher Scientific, Reinach, Switzerland) and 0.5 g/l L of cysteine-HCl (Sigma, Buchs, Switzerland). All media were placed into 100 mL batch culture vessels and maintained under an atmosphere of oxygen-free nitrogen gas by continuously sparging with oxygen-free nitrogen (15 mL/min). The vessel was magnetically stirred, and pH was maintained using a pH controller (Electrolab Biotech Limited, Gloucester, UK). The medium was allowed to equilibrate overnight before fecal inoculation. Batch cultures ran under anaerobic conditions for a period of 24 h during which samples (5 mL) were collected at times of 0 h, 6 h, 18 h and 24 h for high-throughput sequencing and metabolomics analysis. T0 samples were taken from the respective vessels under operating conditions and not from volunteer fecal slurries. Samples were stored at −80 °C until needed. Twenty-four hours is the typical maximum incubation time for batch systems when simulating the large intestine of monogastric animals [59].

### 4.8. Sequencing

DNA extraction was performed using the QIAamp Fast DNA Stool Mini Kit (Qiagen, Hombrechtikon, Switzerland) following the manufacturer’s protocols. Further sample preparation and sequencing was outsourced and performed at the Swiss Integrative Center for Human Health (SICHH; Fribourg, Switzerland). DNA concentrations for the PCR were normalized to 5 ng/µL. Sequencing libraries were created from 2 ng/µL of DNA after the PCR and after applying the 16S Metagenomic Sequencing Library protocol (Illumina, Zürich, Switzerland), targeting the variable V3 and V4 regions of the 16S rRNA gene (a single amplicon of approximately 460 bp) [61]. All 27 samples (plus 1 negative control) were sequenced in one Illumina MiSeq run at a 2 × 300 bp paired-end read length, which yielded between 344,848 (103 Mb) and 842,725 pass-filtered reads (252 Mb) per sample, reaching a median pass-filtered sequencing depth of 160 Mb per sample. Illumina conversion software bcl2fastq2 version v2.18.0.12 was automatically run through the MiSeq local run manager set with default parameters to trim Illumina adapters, demultiplex samples based on their respective index, and generate fastq files. Sequencing reads were evaluated for quality and adaptor contamination using FastQC v0.11.5 [62].

Paired-end sequencing reads were uploaded on the microbiota sequencing and analysis platform imngs2 (https://www.imngs2.org, accessed on 6 July 2022). Analyses were performed using the Rhea pipeline in R [63]. Taxonomy assignment, alpha diversity, and beta diversity based on the generalized UniFrac were generated using the Rhea pipeline. The generalized UniFrac was used to avoid analytical sensitivity to rare and dominant OTUs [64]. The dendrogram of all the samples was calculated by hierarchical clustering using the Ward’s minimum variance method [65]. Beta diversity visualization was performed by unconstrained, non-metric multidimensional scaling (NMDS) [66]. A permutational multivariate analysis of variance using distance matrices (vegan: adonis) was performed to determine if the separation of groups is significant. To aid visualization, replicate samples were averaged and displayed together.

#### Nucleotide Sequence Accession Numbers

Metagenome raw sequencing datasets have been made public through the Sequence Read Archive of the National Center for Biotechnology Information under SRA Accession Numbers SRR8695500 to SRR8695526, and gathered together under the Bioproject Accession Number PRJNA525885 and the Study Accession Number SRP187837 [67].

### 4.9. Volatile Fatty Acid Analysis

For sample analysis, 1 mL aliquots of culture were taken at inoculation (T0), after 6 h (T6), after 18 h (T18), and after 24 h (T24). Samples were centrifuged (13,000× *g*) for 10 min and the supernatant filtered through a sterile 0.45µm syringe filter. The filtrate was injected directly into an HPLC (Agilent, Basel, Switzerland) using a Cation H+ pre-column and Rezex ROA-Organic Acid H+ (8%) column (30 × 4.6 mm, Phenomenex, Aschaffenburg, Germany), which was ran with an isocratic mobile phase of 5 mM H_2_SO_4_ at a flowrate of 0.5 mL min^−1^ (62 min, 70 °C). Lactic acid, acetic acid, succinic acid, propionic acid, and butyric acid were detected by refractive index and UV (210 nm) spectrometry. To aid in visualization, replicate samples were averaged and displayed together.

### 4.10. Targeted Analysis of Lignans, Cyclolinopeptides, and Bile Acids

For the targeted analysis, 100 µL of digested media was evaporated using an Eppendorf concentrator. The dried sample was dissolved in 100 µL of 80% acetonitrile by sonication for 10 min and vigorous shaking for 5 min, followed by sonication for 10 min. The samples were centrifuged at 3000 rpm for 10 min and the clear supernatant was used for mass spectrometric analysis. An API 5500 QTrap mass spectrometer (Sciex, Darmstadt, Germany) coupled to an ExionLC UPLC was used for the detection of lignans, cyclolinopeptides, and bile acids in multiple reaction monitoring mode (MRM). The reference materials for lignans and bile acids were commercially available. Cyclolinopeptides were isolated according to Frank et al. [68]. Data acquisition was performed with Analyst 1.7.0 software (Sciex), data analysis with MultiQuant 3.0.3 (Sciex), and statistical analysis with Metaboanalyst [69]. Mass spectrometric settings for the MRM transitions of lignans and cyclolinopeptides are shown in Table 3.

#### 4.10.1. Lignans and Cyclolinopeptides

Chromatographic separation was carried out on a 100 mm × 2.1 mm i.d., 5 μm, Kinetex XB-C18 column (Phenomenex, Germany, Aschaffenburg) using a linear binary gradient at a column temperature of 40 °C. The injection volume was 1 μL, and the autosampler was cooled to 15 °C. The flow rate was 300 μL/min. Solvent A was 0.1% formic acid, and solvent B was 0.1% formic acid in acetonitrile. The UPLC was programmed using a gradient from 5% B that was held for 0.5 min to 100% B at 10 min. After flushing the column for 5 min at 10 0% B, the column was equilibrated to the starting conditions.

For electrospray ionization, the ion voltage was set to +5500 V in positive mode and to −4500 V in negative mode, and nitrogen was used as the curtain gas (35 psi). Zero-grade air was used as a nebulizer gas (45 psi), and as a drying gas (55 psi), heated to 400 °C. For quantitation, the multiple reaction monitoring mass spectrometry (MRM) transitions were optimized for each reference compound using standard material. Cyclolinopeptides were measured in positive ionization mode and SECO, SDG, ED, EL, (+)-LA, PINO, and PDG were measured in negative ionization mode. The MRM transitions of lignans and cyclolinopeptides are shown in Table 2.

#### 4.10.2. Bile Acids

Bile acid analysis was performed according to Reiter et al. [70]. An electrospray ion voltage of −4500 V and the following ion source parameters were used: curtain gas (35 psi), temperature (450 °C), gas 1 (55 psi), gas 2 (65 psi), and entrance potential (−10 V). For separation of the analytes, a 100 × 2.1 mm, 100 Å, 1.7 μm, Kinetex C18 column (Phenomenex) was used. Chromatographic separation was performed with a constant flow rate of 0.4 mL/min using a mobile phase consisting of water (eluent A) and acetonitrile/water (95/5, *v*/*v*, eluent B), both of which contained 5 mM ammonium acetate and 0.1% formic acid. The gradient elution started with 25% B for 2 min, increased at 3.5 min to 27% B, and then increased in 2 min to 35% B. This was held until 10 min of elution time was reached. Afterwards, the eluent B concentration was increased in 1 min to 43% B, which was held for 1 min before again increasing eluent B in 2 min to 58% B. This concentration of eluent B was held for 3 min isocratically. Finally, the concentration of B was increased to 65% at 17.5 min, with another increase to 80% B at 18 min, following an increase at 19 min to 100% B which was held for 1 min. After a total runtime of 20.5 min, the column was equilibrated to the starting conditions for 4.5 min. The injection volume for all samples was 1 μL, the column oven temperature was set to 40 °C, and the auto-sampler was kept at 15 °C. Data acquisition and instrumental control were performed with Analyst 1.7 software (Sciex, Darmstadt, Germany). The data were analyzed with MultiQuant 3.0.3 (Sciex, Darmstadt, Germany) and Metaboanalyst [69].

## 5. Conclusions

In the study presented here, we compared and examined the changes in microbiota composition and changes in bioactive components of whole flaxseed and defatted flaxseed press cake during microbiota fermentation in vitro. We observed that press cake and whole flaxseed are equivalent in microbiota changes and functionality. However, press cake may be superior as a functional additive in a variety of foods in terms of consumer acceptance as it would be more resistant to oxidative changes, improving sensory perception and storage. Considering that flaxseed press cake is a cheap by-product of flaxseed oil production, the addition of this waste product to consumer foods may yield a sustainable and resource-efficient source of food additives. However, more in vivo studies are needed to ascertain the dysbiotic effect of flaxseed supplementation seen here to see if they pose any dangers to either healthy individuals or people with circulatory diseases.

## Figures and Tables

**Figure 1 ijms-23-10473-f001:**
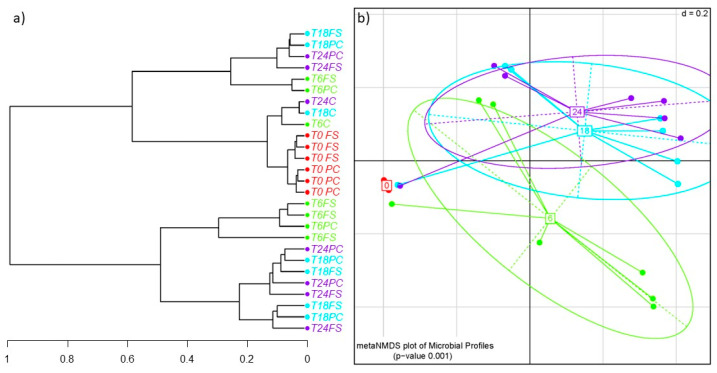
(**a**) Dendrogram and (**b**) generalized Unifrac NMDS plot of human gut microbiota batch cultures observed over 24 h (T0, T6, T18, T24 = observed timepoints 0 h (red), 6 h (green), 18 h (blue), and 24 h (purple); C = control fermentation without flaxseed, PC = digested flaxseed press cake, FS = digested milled whole flaxseed).

**Figure 2 ijms-23-10473-f002:**
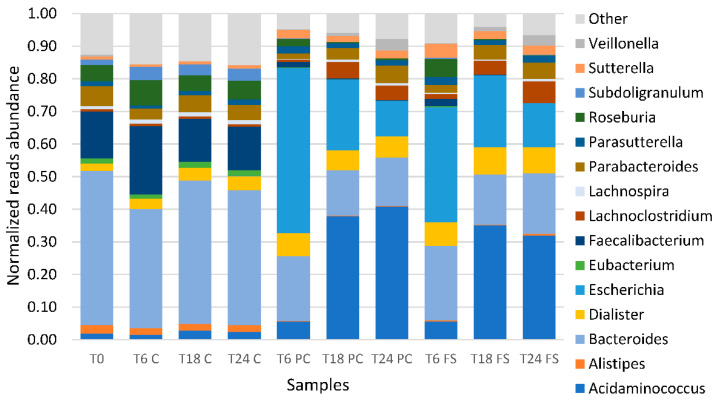
Stacked bar plot of the relative abundance of the 15 most predominant genera of human gut microbiota batch cultures. T0, T6, T18, T24 = observed timepoints 0 h, 6 h, 18 h, and 24 h; C = control fermentation without flaxseed, PC = digested flaxseed press cake, FS = digested milled whole flaxseed.

**Figure 3 ijms-23-10473-f003:**
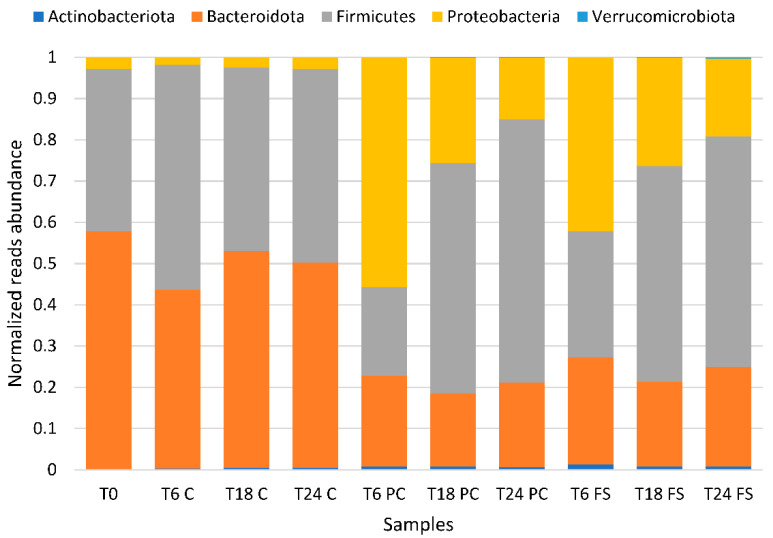
Stacked bar plot of the relative abundance of phyla present in human gut microbiota batch cultures. T0, T6, T18, T24 = observed timepoints 0 h, 6 h, 18 h, and 24 h; C = control fermentation without flaxseed, PC = digested flaxseed press cake, FS = digested milled whole flaxseed.

**Figure 4 ijms-23-10473-f004:**
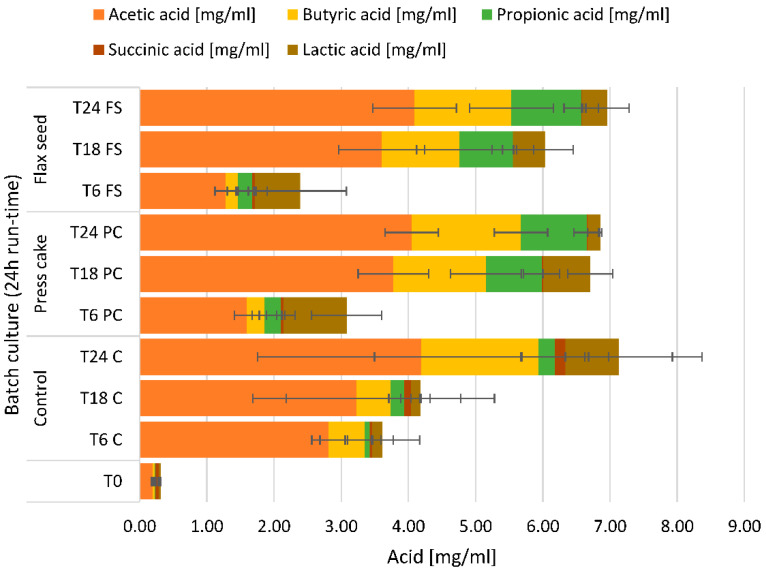
Stacked bar plot of concentrations of volatile fatty acids (±SD). T0, T6, T18, T24 = observed timepoints 0 h, 6 h, 18 h, and 24 h; C = control fermentation without flaxseed, PC = digested flaxseed press cake, FS = digested milled whole flaxseed.

**Figure 5 ijms-23-10473-f005:**
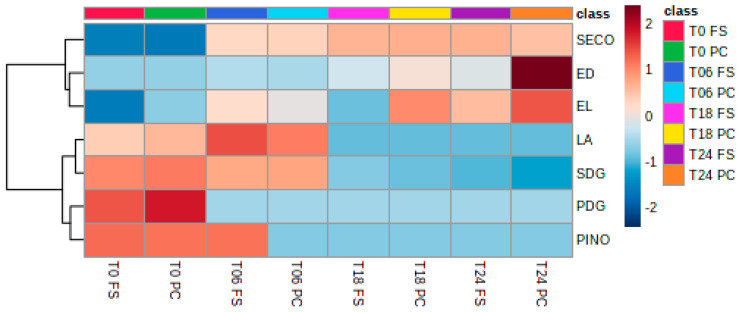
Heatmap of log-transformed concentrations of lignans and their degradation products (Euclidean distance measures and Ward’s clustering algorithm used). T0, T6, T18, T24 = observed timepoints 0 h, 6 h, 18 h, and 24 h; PC = digested flaxseed press cake, FS = digested milled whole flaxseed.

**Figure 6 ijms-23-10473-f006:**
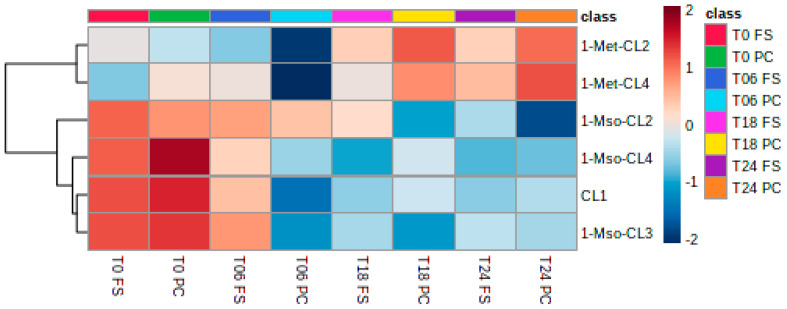
Heatmap of the metabolism of cyclolinopeptides during fermentation. T0, T6, T18, T24 = observed timepoints 0 h, 6 h, 18 h, and 24 h; PC = digested flaxseed press cake, FS = digested milled whole flaxseed.

**Figure 7 ijms-23-10473-f007:**
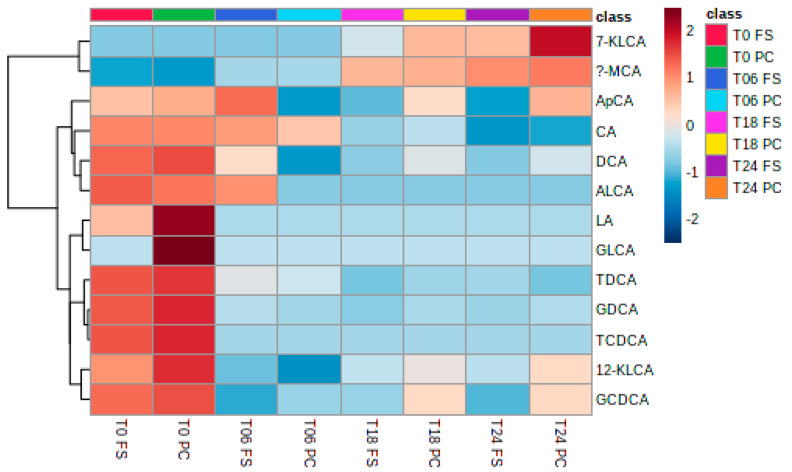
Heatmap of the bile acid production during fermentation. T0, T6, T18, T24 = observed timepoints 0 h, 6 h, 18 h, and 24 h; PC = digested flaxseed press cake, FS = digested milled whole flaxseed.

**Table 1 ijms-23-10473-t001:** Alpha diversity changes of human gut microbiota batch cultures observed over 24 h (T0, T6, T18, T24 = observed timepoints 0 h, 6 h, 18 h, and 24 h; C = control fermentation without flaxseed, PC = digested flaxseed press cake, FS = digested milled whole flaxseed).

	Richness	Normalized Richness	Effective Richness	Shannon Index	Shannon Effective	Simpson Index	Simpson Effective	Evenness
T0	78	70	51	3.28	26.58	0.08	13.07	0.52
C T6	77	71	56	3.35	28.38	0.07	14.22	0.53
C T18	80	74	55	3.29	26.74	0.08	12.53	0.52
C T24	80	72	54	3.28	26.05	0.08	12.36	0.52
T6 PC	71	57	29	2.21	9.98	0.28	4.27	0.36
T18 PC	73	50	34	2.69	15.99	0.14	8.27	0.43
T24 PC	75	52	36	2.75	16.58	0.15	7.68	0.44
T6 FS	79	66	37	2.65	16.82	0.20	7.95	0.42
T18 FS	75	51	33	2.59	14.60	0.16	7.70	0.42
T24 FS	73	50	33	2.78	16.57	0.12	9.39	0.45

**Table 2 ijms-23-10473-t002:** Continuous culture growth medium (CMGM, g liter^−1^ in distilled water) [58,60].

Component	g × L^−1^
Starch (BDH Ltd.)	5.0
Pectin (citrus)	2.0
Guar gum	1.0
Mucin (porcine gastric type III)	4.0
Xylan (oat spelt)	2.0
Arabinogalactan (larch wood)	2.0
Inulin	1.0
Casein (BDH Ltd.)	3.0
Peptone water	5.0
Tryptone	5.0
Bile salts No. 3	0.4
Yeast extract	4.5
FeSO_4_ × 7H_2_O	0.005
NaCl	4.5
KCl	4.5
KH_2_PO_4_	0.5
SO_4_ × 7H_2_O	1.25
CaCl_2_ × 6H_2_O	0.15
NaHCO_3_	1.5
Cysteine	0.8
Hemin	0.01
Tween 80	1.0

**Table 3 ijms-23-10473-t003:** Mass spectrometric settings for the MRM transitions of lignans and cyclolinopeptides. Q stands for quantifier and ID stands for identifier. A terminology showing the structural relations of the cyclolinopeptides was used: CL1 → CLA, 1-Met-CL2 → CLB, 1-Mso-CL2 → CLC, 1-Met-CL3 → CLD’, 1-Mso-CL3 → CLD, 1-Met-CL4 → CLE’, 1-Mso-CL4 → CLE.

Sample Name	Polarity	Q1 (Da)	Q3 (Da)	Dwell Time (ms)	DP (V)	EP (volts)	CE (volts)	CXP (volts)
(+)-Lariciresinol Q	-	359.027	175.000	25	−115	−10	−32	−9
(+)-Lariciresinol ID	-	359.027	160.400	25	−115	−10	−40	−7
Enterodiol Q	-	301.031	106.500	25	−140	−10	−42	−15
Enterodiol ID	-	301.031	270.900	25	−140	−10	−34	−11
Enterolactone Q	-	296.998	106.500	25	−100	−10	−40	−5
Enterolactone ID	-	296.998	121.100	25	−100	−10	−32	−1
Secoisolariciresinol diglucoside ID	-	685.153	361.000	25	−290	−10	−56	−15
Secoisolariciresinol diglucoside Q	-	685.153	58.900	25	−290	−10	−112	−9
Secoisolariciresinol Q	-	361.041	165.100	25	−40	−10	−34	−5
Secoisolariciresinol ID	-	361.041	120.900	25	−40	−10	−58	−7
Pinoresinol diglucoside Q	-	682.108	520.100	25	−210	−10	−22	−21
Pinoresinol diglucoside ID	-	682.108	151.100	25	−210	−10	−66	−5
Pinoresinol Q	+	359.067	137.100	25	+96	+10	+33	+6
Pinoresinol ID	+	359.067	175.100	25	+96	+10	+23	+6
CL1 Q	+	1040.600	941.4	25	+296	+10	+41	+36
CL1 ID	+	1040.600	828.4	25	+296	+10	+49	+30
1-Met-CL2 Q	+	1058.6	945.4	25	+296	+10	+45	+36
1-Met-CL2 ID	+	1058.6	588.2	25	+296	+10	+61	+20
1-Mso-CL2 Q	+	1074.5	961.3	25	+286	+10	+49	+38
1-Mso-CL2 ID	+	1074.5	489.1	25	+286	+10	+73	+18
1-Met-CL3 Q	+	1048.5	935.3	25	+241	+10	+35	+34
1-Met-CL3 ID	+	1048.5	822.2	25	+241	+10	+45	+30
1-Mso-CL3 Q	+	1064.5	951.3	25	+266	+10	+41	+36
1-Mso-CL3 ID	+	1064.5	1000.4	25	+266	+10	+37	+36
1-Met-CL4 Q	+	961.5	814.3	25	+296	+10	+35	+30
1-Met-CL4 ID	+	961.5	358.1	25	+296	+10	+57	+40
1-Mso-CL4 Q	+	977.5	358.1	25	+296	+10	+59	+12
1-Mso-CL4 ID	+	977.5	211.1	25	+296	+10	+67	+24

## Data Availability

Metagenome raw sequencing datasets are available at the National Center for Biotechnology Information under the SRA Accession Numbers SRR8695500 to SRR8695526, and gathered under the Bioproject Accession Number PRJNA525885 and the Study Accession Number SRP187837.

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
