# Peer review of "Dietary Modulation of the Human Gut Microbiota and Metabolome with Flaxseed Preparations"

_ijms, 2022, doi:10.3390/ijms231810473_

Round 1

Reviewer 1 Report

1. This is an interesting study and a valuable addition to the literature on the topic of flaxseed and its effects on the bacterial microbial environment. My biggest concern is a paucity of discussion as to how these results differ or are similar to previous studies of flaxseed ingestion and microbiome effects. The results need to be put into more of a context with existing data on the topic. The authors need to refer to past work directly relevant to the present results. I realize the papers cited below are very new but the authors should have been aware of them and discuss them in the Discussion section. It is not as if nothing has been done on the topic as the authors tend to give the impression on lines 67 and 68 of the Introduction. The relevant studies are found in: 

Newman et al, Am J Physiol Cell Physiol; Taibi et al J Nutr Biochem 98:108818, 2021; Sun et al J Sci Food Agric may 14, 2022; Taibi et al Data Brief 38:107409, 2021; Resch et al Microorganisms 9:1037, 2021.

2. The first paragraph of the Introduction is unnecessarily negative. The toxicity of flaxseed is emphasized along with other unwanted side effects. There is no need to include this when it is not at all central or a focus of this study. Please remove most of that first paragraph.

3. One of the major limitations of the study is its use of an in vitro batch culture model. The authors do describe stool sample collection from humans who ingested flaxseed but it is unclear in the Methods section how much they ate, in what form, with what other dietary components, and for how long before testing the stools. If it was simply one ingestion period, this is unlikely to give reliable saturation results as one would obtain after >1 week of ingestion. This detail needs to be included. In addition, the limitations of this (and for in vitro results as opposed to the actual ingestion and digestion of the flaxseed) needs to be discussed in the Discussion.

4. It is important to recognize that defatted flaxseed still contains a significant amount of fat, most notably ALA. Unless the authors have actually produced a fat deficient sample, the contaminating action of 10% ALA (that is typically found in defatted flaxseed) is still substantial and would have a major confounding effect on their interpretation of the results.

Author Response

We appreciate the editor’s reviewer’s comments and believe to have answered the comments sufficiently below for the paper to be reconsidered for publication:

  1. This is an interesting study and a valuable addition to the literature on the topic of flaxseed and its effects on the bacterial microbial environment. My biggest concern is a paucity of discussion as to how these results differ or are similar to previous studies of flaxseed ingestion and microbiome effects. The results need to be put into more of a context with existing data on the topic. The authors need to refer to past work directly relevant to the present results. I realize the papers cited below are very new but the authors should have been aware of them and discuss them in the Discussion section. It is not as if nothing has been done on the topic as the authors tend to give the impression on lines 67 and 68 of the Introduction. The relevant studies are found in: 

Newman et al, Am J Physiol Cell Physiol; Taibi et al J Nutr Biochem 98:108818, 2021; Sun et al J Sci Food Agric may 14, 2022; Taibi et al Data Brief 38:107409, 2021; Resch et al Microorganisms 9:1037, 2021.

  • The articles were integrated into the manuscript, where necessary.
  1. The first paragraph of the Introduction is unnecessarily negative. The toxicity of flaxseed is emphasized along with other unwanted side effects. There is no need to include this when it is not at all central or a focus of this study. Please remove most of that first paragraph.
  • The negative side effects have been removed from the first paragraph.
  1. One of the major limitations of the study is its use of an in vitro batch culture model. The authors do describe stool sample collection from humans who ingested flaxseed but it is unclear in the Methods section how much they ate, in what form, with what other dietary components, and for how long before testing the stools. If it was simply one ingestion period, this is unlikely to give reliable saturation results as one would obtain after >1 week of ingestion. This detail needs to be included. In addition, the limitations of this (and for in vitro results as opposed to the actual ingestion and digestion of the flaxseed) needs to be discussed in the Discussion.
  • The study used fecal samples from individuals on a standard western diet that did not consume flaxseed products. Hence the batch culture model was used to examine the effect of flaxseed supplementation on the fecal microbiota in vitro only.
  1. It is important to recognize that defatted flaxseed still contains a significant amount of fat, most notably ALA. Unless the authors have actually produced a fat deficient sample, the contaminating action of 10% ALA (that is typically found in defatted flaxseed) is still substantial and would have a major confounding effect on their interpretation of the results.
  • The flaxseed preparations still contained residual amounts of fat. However, literature shows that flaxseed fibers influence the production of enterolactones by gut microbiota and not ALA. In addition, we compared the flaxseed press cake with whole, milled flaxseed and thus any confounding effects are negligible for this publication.

Author Response

We appreciate the reviewer’s comments and believe to have answered the comments sufficiently below for the paper to be reconsidered for publication:

The article titled Dietary Modulation of the Huma Gut Microbiome and Metabolome with Flaxseed Preparations addresses the goal of the special issue for the International Journal of Molecular Sciences Bioactive molecules and prebiotics for gut health and beyond. The authors choose an attractive dietary component that has gained popularity in the last years for the health benefits highlighted in this paper. Therefore, I would suggest accepting the manuscript with some revisions before publication. Please, find below comments:

  1. Across the document, gut microbiota and gut microbiome are used interchangeably; however, they refer to two different things. Please, use the name that most accurately describes your data. I would suggest using gut microbiota rather than gut microbiome since there is no pathway analysis, just descriptive analysis of the gut microbes.

Microbiome has been replaced with microbiota throughout the manuscript

  1. Please update in vitro to italic across the document.

in vitro” has been updated throughout the document.

  1. The abstract would benefit from clearly staying the aim of this project as it is described in the discussion lines 291-294, reducing the verbiage for the introduction and providing more characters to describe the methods better. Between lines 26-28 all methods are described in one sentence making the reader hard to follow. Please, consider updating.

Unfortunately, due to character constraints for the abstract, an expanded explanation of the methods is impossible.

  1. Line 52-53 flaxseeds benefits are listed, but only one reference is mentioned. You might want to consider rearranging this paragraph with information from the discussion section.

This has been updated and references have been added.

  1. Line 66 This flaxseed polysaccharide is referring to arabinoxylan or acidic rhamnose?

It’s a heterogeneous polysaccharide consisting of arabinoxylan and acidic rhamnose as described in lines 66-67.

  1. Line 124-25 Please update the aim as described in lines 291-294. The last paragraph of the introduction does not describe what is described in the results section.

The aims have been rewritten accordingly.

  1. Line 154-155 are not required. This describes the methods. Results should clearly state the results from the analysis.

The lines have been deleted.

  1. Figure 1. The color codes for the different groups are not explained at the foot of the figure. Please, add a description.

Color codes have been added in the figure legend.

  1. Figures for the relative abundance will benefit from adding more space between each group. The space between T24 and T6 bars.

The gap between each group has been increased.

  1. Why did you not report the quantity that the SCFAs increased? Why is there not a statistical analysis to assess if there is or is not a significant change (increase or decrease) in the SCFAs? This might be of interest as the title, and the aim of the study is to determine the impact of the flaxseed fiber on gut microbiota and metabolome.

Changes in SCFAs and the statistical significance of changes has been described in lines 234-251.

  1. Same comment for the Bile Acids, why not compare the change in the secondary bile acids among the different treatments? Why did you not relate the GM changes and BA changes? A correlation analysis might be of interest for future researchers that want to make a case to test causation rather than association. I think the paper would benefit from such additional analysis.

A comment on bile acid production in this study has been added (lines 402-417).

  1. Section 3.2 discussion between lines 317-325, all the benefits for SCFA are stated, but none of those benefits are tested in this article. I think for discussion should be a comparison of what is out there on the topic that you are testing (i.e., flaxseeds or other fibers on SCFA synthesis). Please, consider revising this section.

We thank the reviewer for this comment. However, we do not see the benefit to reproduce literature data that is commonly accepted in the scientific community.

  1. Lines 601-602 it is not describing the goal of this study. Please consider updating.

The beginning of the conclusion has been updated to better align with the goal of this study.

  1. I would suggest addressing the limitations of the techniques utilized in this paper (e.g., in vitro model vs. in vivo). Further description of the introduction and/or discussion addressing other literature that used the technique will help to understand why it is important to use this technique first over animal models or clinical trials.

We thank the reviewer for the comments. We have added further literature that underlines the importance of in-vitro trials for generation of physiologically relevant metabolites and identification of microbiome populations in response to food constituents. We would like to emphasize that in vitro models are the easiest way to generate physiologically relevant data without the need for statistically relevant human cohorts and the ethical documentation required to conduct a human trial.